# Moisture Transport of Axial-Compression-Damaged Mortar and Concrete in Atmospheric Environment

**DOI:** 10.3390/ma15165498

**Published:** 2022-08-10

**Authors:** Yong Zhou, Weiping Zhang, Fei Tong, Xianglin Gu

**Affiliations:** 1Key Laboratory of Performance Evolution and Control for Engineering Structures of Ministry of Education, Tongji University, 1239 Siping Road, Shanghai 200092, China; 2Department of Structural Engineering, Tongji University, 1239 Siping Road, Shanghai 200092, China; 3Basic Construction Management Department, Hangzhou International Airport Co., Ltd., Hangzhou 311207, China

**Keywords:** moisture transport, microcracks, pore size distribution, damage-representative radius

## Abstract

The moisture transport of axial-compression-damaged mortar and concrete was experimentally and analytically studied in this paper. Five stress levels, i.e., 25%, 40%, 55%, 70%, and 85%, of the corresponding ultimate compressive strengths were selected for mortar and concrete specimens with the water cement ratio (w/c) of 0.5. Porosities and sorptivities of mortar or concrete before and after axial compression were measured and compared. Based on the Lucas–Washburn equation on absorption, the relationship between sorptivity and pore size distribution as well as porosity was established. A damage-representative radius was proposed to simply quantify the variation of pore characteristics of damaged mortar and concrete, and the moisture transport of axial-compression-damaged mortar and concrete could be predicted by summing the contributions to water absorption from the original pore system and the pore-equivalent microcrack system. It is shown that the porosities of mortar and concrete only slightly increase with the damage level, but the sorptivities are sensitive to axial compression damage, i.e., increasing nearly monotonically with the stress level from 0.3326 to 0.3533 mm/min^0.^5 for damaged mortar specimens (w/c = 0.5) and from 0.1970 to 0.2226 mm/min^0.5^ for damaged concrete specimens (w/c = 0.5). The increase trend became more apparent for both materials after a threshold of 40–55% of the corresponding ultimate compressive strengths, which is within the service load of structures, indicating that damage should be considered for chloride ions and water transport in concrete in the tidal zone. The predicted moisture diffusivities of damaged mortar and concrete show marginal difference from those of sound materials because the damage-representative radius could be underestimated due to elastic recovery of materials after unloading.

## 1. Introduction

The degradation mechanisms of concrete structures mainly include freeze–thaw damage, carbonation, alkali–silica reaction, and mechanical-load-induced damage for concrete and corrosion for reinforcement. Among all the mechanisms, moisture, including liquid water and water vapor, either acts as the medium to transport aggressive agents into concrete or directly participates in the degradation process as a reactant. Therefore, establishing an accurate analytical model and determining its characteristic parameters related to moisture transport are essential for the life-cycle design of concrete structures.

In atmospheric environments, the mechanism of moisture ingress into unsaturated concrete greatly depends on the boundary conditions of the structure. When concrete is in a high-humidity environment, the ingress of water vapor from the air into the concrete can be recognized as moisture adsorption, and the moisture diffusivity, $D_{h}$, is used to describe the speed of the transport process. However, if concrete is in direct contact with water, the process of water moving into the unsaturated concrete under ambient pressure or capillary pressure should be recognized as water absorption, and the speed of the water intake is called sorptivity, $S_{0}$, which reflects the suction ability of porous materials. The moisture transport in cementitious materials is closely related to their pore characteristics, no matter whether the mechanism is adsorption or absorption.

With the assumed relationship between the moisture diffusivity and relative humidity or water content in undamaged cementitious materials, many empirical models for moisture diffusivity in undamaged cementitious materials have been proposed [1,2,3,4,5]. The disadvantages of the empirical models are obvious. Since the empirical models are from the fitting of experimental data, they greatly depend on the accuracy of the data. However, different experimental methods will give significantly different results and limited data cannot fully reflect the moisture transport properties of highly stochastic materials [6]. Thus, more physically based moisture transport models have been developed by further studying transport mechanisms and composite theory. For example, to consider the influence of saturation degree and microstructural characteristics, Zhang et al. [7] introduced two empirical parameters, i.e., the relative permeability of water and the hindrance diffusion coefficient of water vapor, into macroscopic transport equations. Maekawa et al. [8], Ishida et al. [9], and Huang et al. [10] first obtained the transport equation for a single cylindrical pore based on the basic theories of thermodynamics, and then integrated all the pores of an assumed pore size distribution to obtain the moisture diffusivity of cementitious materials. These models are more theoretical, since they consider the transport mechanism at the pore scale; however, the question of how to obtain an accurate pore size distribution is the key problem to be solved. The commonly used solutions include mercury intrusion porosimetry (MIP) and some simple distribution functions, e.g., the Raleigh–Ritz distribution function. Based on the Barrett–Joyner–Halenda method [11], Zhang et al. [12] theoretically related the pore size distribution to the water–cement ratio (w/c) of cement paste by analyzing the water adsorption isotherms and assuming a simplified formula for the thickness of the adsorbed layer. The moisture diffusivity and humidity response of cement paste and concrete were obtained by integrating the calculated pore size distribution with the consideration of the hysteresis effect and composite theory.

During their service stage, cementitious materials are inevitably subjected to some unfavorable actions, such as external loading, drying shrinkage, and thermal gradients, etc., which may produce microcracks inside the materials. The microcracks can be the development of initial capillary pores, new nucleation in the cement paste, or an interfacial transition zone (ITZ). Superposition of the microcracks onto the original pore network will enhance the mass-transport performance of the materials and impair the durability of structures. Ghasemzadeh and Pour-Ghaz [13] proved that the sharp-front theory was not suitable for the absorption prediction of freeze–thaw-damaged concrete and proposed a conceptual model in which distributed cracks were assumed as uniformly distributed parallel plates of equal thickness, and composite theory was adopted to calculate the sorptivity of damaged concrete from the sorptivities of undamaged concrete and cracks. Wang and Zhang [14] modified this conceptual model by replacing plate cracks with uniformly distributed capillary tubes of identical pore radius and assumed that the growth of microcracks was completely attributable to the increment of capillary tube radius.

Many experimental investigations have been carried out to understand the influence of microcracks on moisture transport in cementitious materials. Microcracks were produced by different methods, such as tension [15,16,17,18,19], compression [20,21,22,23,24,25,26,27,28,29,30,31], flexure [32], and freeze–thaw cycles [16,30,32]. Mass-transport properties were measured by different testing methods, such as chloride permeability [26,27,29,30,31,32,33,34], sorptivity [16,17,28,31,34], gas permeability [22,23,24,25,29,34], and water permeability [18,19,20,21,22,34]. However, not many of these studies directly addressed the effect of microcracks on unsaturated transport. Wang and Li [17] researched the sorptivity of concrete damaged by uniaxial tension or compression of up to 90% of the corresponding ultimate strengths and observed that the damage significantly increased the sorptivity of concrete. Zhou et al. [28] studied the crack geometry, as quantified by density, length, orientation, and connectivity, in concretes subjected to cyclic axial compression. They related the variation of transport properties to the index of volumetric crack density and found a maximum 30% increase in sorptivity for ordinary concrete based on the measurement during a very short period of 36 min. Ghasemzadeh et al. [34] compared the sensitivity of freeze–thaw-induced microcracks to different mass-transport measurement methods, i.e., electrical resistivity, rapid chloride permeability, sorptivity, drying, air/water permeability, and desorption isotherm. They found that water/air permeability and initial sorptivity increased exponentially and linearly with microcracks, respectively, and the chloride permeability test and electrical resistivity were only sensitive at low damage levels. Liu et al. [19] experimentally concluded that the permeability of cracked, engineered cementitious composites under tension depended more on the crack than the crack pattern. Tran et al. [31] measured the water permeability and chloride diffusivity of concrete under sustained compressive stress and found a slight decrease in transport properties of the researched concretes when the compressive stress is smaller than 40% of the ultimate compressive strength and a quick increase above 40% of the ultimate compressive strength.

This paper proposed an analytical method to predict the moisture transport in unsaturated mortar and concrete during their service stage. The method assumed that axial compression induced the microcracks as a new pore system was superimposed on the original pore system in sound mortar and concrete. The characteristics of the new pore system were inferred from water-absorption experiments. Then, the moisture transport of axial-compression-damaged mortar and concrete could be calculated by combining the moisture transport in the two pore systems. This method shows the correlation of the extent of the stress-induced damage with the variation of the pore characteristics; that is, it can calibrate the damage level of cementitious materials through the sorption test data on the one hand, and, on the other hand, can predict the moisture transport in unsaturated damaged cementitious materials once the pore characteristics are known. This paper is organized as follows. Section 2 and Section 3 report the details of the experimental scheme and the experimental results, respectively. The porosity, pore size distribution, and the sorptivity of damaged and undamaged mortar and concrete, which were the necessary inputs for the analytical model, were obtained. Section 4 shows the formulation and application of the analytical model, in which the method of determining the damage-representative radius was introduced. Finally, some concluding remarks and further research are presented. 

## 2. Materials and Methods

Ordinary Portland cement with an apparent density of 3077 kg/m^3^ was used. Fine aggregates were well-graded river sands with the fineness modulus of 2.5–2.6 and an apparent density of 2604 kg/m^3^ (Figure 1). The upper and lower bounds of grading zone II were adopted from GB/T 14684 [35]. Crushed granite stones, grain size groups 4.79–9.50 mm and 9.50–16.0 mm, mixed in the ratio of 55%:45%, served as the coarse aggregates with an apparent density of 2628 kg/m^3^. Tap water was used during the mixing and casting.

Three groups of specimens, i.e., cement paste, mortar, and concrete, were fabricated with three w/cs of 0.4, 0.5, and 0.6 for each group. Table 1 shows the mixture proportions and the number of specimens used for different purpose. All specimens had a prismatic shape, but the dimensions were 70.7 mm × 70.7 mm × 230 mm for the cement paste specimens and mortar specimens and 100 mm × 100 mm × 300 mm for the concrete specimens. This difference was due to the consideration of different aggregate sizes in the mortar and concrete specimens. The specimens were cured for 28 days as standard before the following steps were carried out.

From each group of cement paste, mortar, and concrete specimens with a given w/c, one specimen was selected randomly and crushed to approximately 1 cm^3^ particles, then about 20 g of particles were sealed with anhydrous calcium chloride in a plastic bag and sent to the Key Laboratory on Civil Engineering Materials at Southeast University, Nanjing, PRC, for an MIP test to obtain the pore size distribution. As shown in Figure 2, the undamaged specimens serving as the reference were cut into 3 (for cement paste and mortar) and 4 (for concrete) 50 mm-thick slices. Experiments with the damaged specimens were only conducted on the mortar and concrete specimens with the w/c of 0.5. The ultimate compressive strengths of mortar and concrete were firstly obtained by averaging the peak stress values of three corresponding specimens under axial compression using the server-controlled universal testing machine SHT4206 (maximum capacity 2000 kN) from the MTS Systems China Co., Ltd. Five compressive stress levels were selected to produce damage in the mortar (concrete) specimens, i.e., 25%, 40%, 55%, 70%, and 85% of the ultimate compressive strength of mortar (concrete). Five mortar (concrete) specimens were axially compressed until their stresses have one-to-one correspondence to the five stress levels, and the compressive loads were maintained for 30 min before unloading to allow the full development of microcracks. The cutting methods for the damaged specimens were similar to those for the undamaged specimens. However, only the 100 mm-length or 150 mm-length central segments of the damaged mortar or concrete specimens were used to obtain the 50 mm-thick slices, because it is believed that damage is more likely to occur in the central part of a compressed specimen.

After being cut and numbered, the slices were polished on their cutting faces. The exact dimensions of the slices were measured by a Vernier caliper to obtain the real volume, V, and sorption area, A. Then, the slices were placed in a drying oven at the temperature of 105 °C ± 5 °C for 24 h, and weighed after they were naturally cooled to room temperature (20–25 °C) while in the oven. The drying-and-cooling procedure was repeated until the difference of the mass, weighed successively, was less than 5% of the smaller value, and the smaller weight was deemed as the dry mass, $m_{d}$. Although some researchers [36] have suggested a moderate degree of oven drying, e.g., 3 days at 50 °C, in case of shrinkage-induced microcracks disguising those caused by mechanical load, 105 °C oven drying was still adopted in this experiment because similar porosity and pore size distribution after 60 °C and 105 °C oven drying were reported by Galle [37] for CEM I cement paste with 0.5, the same w/c that was used in this paper. For both the undamaged and damaged specimens, two slices of the same specimen were selected randomly to measure the sorptivity first then the porosity, while the others were only used for the porosity measurement. Before the sorption experiment, five faces except the sorption face of the dry slices were sealed with paraffin wax to ensure the one-dimensional transport of water. After the wax was solidified, the wax-sealed slices were placed in a vacuum oven to prevent possible adsorption of the moisture in the air, and they were weighed before the sorption experiment. The dry mass of the wax-sealed slices was denoted as $m_{sd}$.

The porosity measurements were obtained as follows: (1) the slices were soaked in a water-filled plastic drum and weighed after being towel-dried every 12 h; (2) the soaking and weighing were repeated until the difference in the mass, weighed successively, was less than 5% of the higher value; (3) the procedure was stopped, and the higher value was taken as the saturated mass, $m_{sat}$. 

The porosity of the wax-sealed slices can be expressed as
(1)$$\Phi =\frac{m_{sat}-m_{sd}}{\rho_{l}V}$$

The porosity of the wax-free slices can be expressed as
(2)$$\Phi =\frac{m_{sat}-m_{d}}{\rho_{l}V}$$
where $\rho_{l}$ represents the density of water at 1000 kg/m^3^.

The weighing method used in this sorption experiment is traditional and widely used. A schematic of the homemade simple device is shown in Figure 3. ABS plastic cylinders, placed parallel to one another, were bonded on the iron pan to maintain a height difference of less than 3–5 mm between the water surface and the soaked faces of the specimens [38].

The timing began once the soaked face was immersed in the water. The mass of each slice, m(t), was recorded at the time intervals specified in ASTM C1585 [39] (1 min ± 2 s, 5 min ± 10 s, 10 min ± 2 min, 20 min ± 2 min, 30 min ± 2 min, 60 min ± 2 min, 120 min ± 5 min, 180 min ± 5 min, 240 min ± 5 min, 300 min ± 5 min, 360 min ± 5 min). The cumulative sorption length, i, was calculated as
(3)$$i=\frac{\Delta m}{A\rho_{l}}=\frac{m\left(t\right)-m_{sd}}{A\rho_{l}}=S_{o}\sqrt{t},$$
where $S_{o}$ is the sorptivity (mm/min^0.5^); Δm is the total water increment at the time t (kg); A is the sorption area (m^2^); $\rho_{l}$ is the density of the water (1000 kg/m^3^). Equation (3) shows a linear relationship between the cumulative sorption length, i, and the square root of time, t; the sorptivity, $S_{o}$, is on the slope of the straight line i-t^0.5^.

## 3. Results

### 3.1. Pore Size Distribution by MIP

The cumulative pore volume fraction curves of cement paste, mortar, and concrete are comparatively shown for different w/cs in Figure 4. It can be seen that the three curves of the same w/c are almost coincident except that of CP0.4. This is in good agreement with the experimental results of Winslow [40]. Therefore, it can be assumed that the introduction of aggregates will not make a significant difference between the pore size distributions of cement paste, mortar, and concrete; thus, the ITZ can be neglected. This assumption is also supported by [41,42]. So, the pore size distribution curve of cement paste can be used to approximately represent those of mortar and concrete with the same w/c. 

### 3.2. Porosities and Sorptivities of Undamaged Specimens

The measured porosities and sorptivities of undamaged specimens are shown in Figure 5 and Figure 6, respectively. The points marked with “–Mean” in the figures represent the mean values of the measured data of one kind of material at a given w/c. It can be seen that both the porosities and the sorptivities increase with the w/c, regardless of the material type. Since the porosity and pore size distribution are influenced by the w/c, and since the sorptivity is closely related to the pore characteristics, the sorptivity varies with the w/c as expected.

When the ITZ is neglected and the porosity of aggregates is taken as zero, the porosities of the cement paste phase contained in mortar or concrete, and the porosities of the mortar phase contained in concrete, can be calculated. These are listed in Table 2. The mean values of the measured porosities and sorptivities are also shown in Table 2. It can be observed that the calculated porosities are very close to the measured ones with an absolute error of less than 8.18%. Hence, for aggregates of low porosity, such as granite, where the ITZ can be ignored, it can be assumed that the aggregates only dilute the porosity rather than changing the pore size distribution curve of the cement paste or mortar.

### 3.3. Porosities and Sorptivities of Damaged Specimens

The measured porosities and sorptivities of damaged specimens are shown in Table 3 and Table 4, respectively. For the slices prefixed by asterisk in both Table 3 and Table 4, the sorption experiment was performed before the porosity measurement, while for those slices without an asterisk, only the porosity measurement was taken. 

Figure 7 and Figure 8 show the measured porosities versus the stress levels for the damaged mortar and concrete specimens, respectively. Except for the damaged mortar specimens, after the stress levels of 25% and 40% of the ultimate compressive strength, the porosities of the damaged specimens were slightly higher than those of the undamaged specimens, indicating that the original pore system slightly develops by the axial compression. The marginal increase in the porosity versus the stress level may be the combined results of several factors, as follows: (1) the irreversible consolidation of cementitious materials decreases the pore volume; (2) the nucleation and development of microcracks when the material stress–strain curve behaves nonlinearly increase the porosity; (3) the elastic recovery after unloading the materials partly or completely closes some microcracks and reduces the measured porosity; (4) the rehydration of the materials during the porosity test may decrease the pore volume, which is probably more significant for damaged specimens since there are more water transport paths. It is believed that factor 3 is the dominant reason, but factor 4 may also play its role.

The measured sorptivities versus the stress levels of damaged mortar and concrete specimens are shown in Figure 9 and Figure 10, respectively. It can be seen that the sorptivities increase with the stress level. Since the direction of the compressive load is parallel to the length and direction of pores, the pores will be radially enlarged and will further develop with the increase in stress level, making the intake of water more convenient. Sorptivity is sensitive to damage, which is proven by its prompt rise with the increase in stress level. Because only marginal difference can be observed between the porosities of damaged and undamaged specimens (Figure 7 and Figure 8), it can be concluded that sorptivity depends on not only porosity but also pore characteristics. It is well known that absorption is one of the major ways for chloride ions and water to penetrate into concrete during a wetting process in the tidal zone, and many studies have shown that reinforced concrete structures in the tidal zone are most likely to suffer chloride-ion-induced corrosion; therefore, damage caused by mechanical loads should be considered in the durability design of reinforced concrete structures in tidal zones.

## 4. Analytical Analysis

### 4.1. Analytical Model of Sorptivity

Based on the Hagen–Poiseuille equation and the Kelvin equation, the liquid level in a single pore can be expressed by the Lucas–Washburn equation [43], as follows:(4)$$y_{r}=\sqrt{\frac{r\sigma }{2\eta }t},$$
where $y_{r}$ is the liquid level in a pore (m); r is the pore radius (m); σ is the surface tension of water with the value of 0.0722 N/m when *T* = 293.15 K; η is the viscosity coefficient of water, which is 9.9541 × 10^−4^ Pa·s when *T* = 293.15 K; t is the transport time (s). The cumulative mass of sorption in the pore, $\Delta m_{r}$, is obtained by multiplying $y_{r}$ with the cross-sectional area of the pore, $\pi r^{2}$, and the water density, $\rho_{l}$, as follows:(5)$$\Delta m_{r}=\rho_{l}\pi r^{2}y_{r}=\pi r^{2}\rho_{l}\sqrt{\frac{r\sigma }{2\eta }t}.$$

Since the pore radius, r, σ, and η are all constants for a given pore, a linear relationship between $\Delta m_{r}$ and t^0.5^ can be observed in Equation (5). For porous media, such as the cementitious materials, pore sizes are continuous and can be set as $\left(r_{min},r_{max}\right)$. The cumulative mass of water intake, ∆*m*, can be calculated by the integration on the real range of pore sizes, that is
(6)$$\Delta m=\sum_{\left(rmin,rmax\right)}\Delta m_{r}=\sum_{\left(rmin,rmax\right)}\pi r^{2}\rho_{l}\sqrt{\frac{r\sigma }{2\eta }t}$$

The areal porosity, $\Phi_{s}$, can be defined as
(7)$$\Phi_{s}=\frac{\sum_{\left(r_{min},r_{max}\right)}\pi r^{2}}{A}=\sum_{\left(rmin,rmax\right)}\frac{\pi r^{2}}{A},\;$$
where A is the water-absorption area. The areal porosity, $\Phi_{s}$, and volumetric porosity, Φ, are related to each other, as shown by the following [7]:(8)$$\Phi =\tau \Phi_{s},$$
where τ is the tortuosity and is equal to $\pi^{2}/4$ for a uniformly random porous medium.

If the pore size distribution function G(r) is known, e.g., by the MIP test, then the fractional pore volume of the pores with radius r is ΦG(r)dr; therefore, the ratio of the pore’s cross-sectional area to the water-absorption area can be calculated as follows:(9)$$\frac{\pi r^{2}}{A}=\frac{\Phi G\left(r\right)}{\tau }dr,$$

Then, the cumulative water sorption per unit area can be rewritten as
(10)$$\frac{\Delta m}{A}=\sum_{\left(rmin,rmax\right)}\frac{\pi r^{2}}{A}\rho_{l}\sqrt{\frac{r\sigma }{2\eta }t}=\rho_{l}\left(\int_{r_{min}}^{r_{max}}\sqrt{\frac{r\sigma }{2\eta }}\frac{\Phi G\left(r\right)}{\tau }dr\right)\sqrt{t}.$$

By comparing Equation (10) with Equation (2), the sorptivity, $S_{o}$, can be given as
(11)$$S_{o}=\frac{\Phi }{\tau }\int_{r_{min}}^{r_{max}}\sqrt{\frac{r\sigma }{2\eta }}G\left(r\right)dr.$$

Substituting the pore size distribution function, G(r), obtained from the MIP test, and the mean values of the measured porosity presented in Section 3 into Equation (11), the sorptivities of cement paste, mortar, and concrete can be computed and compared with the measured values in Table 5. It should be noted that Equation (11) is obtained by assuming that the pores are parallelly distributed, which is different from a real pore system, so only the pores with a radius smaller than 50 nm have been considered in the calculation to avoid exaggerating the sorptivity. The calculated sorptivities are very close to the measured ones, with less than 10% absolute errors; therefore, Equation (11) can be used to obtain the sorptivity of cementitious materials after the pore size distribution function, G(r), and porosity have been given.

### 4.2. Damage-Representative Radius

The damage evolution of loaded cementitious materials is a process of the nucleation, propagation, and interconnection of internal microcracks until macrocracks appear on specimen surfaces. This process changes the pore characteristics of cementitious materials, such as porosity, pore size distribution, and pore connectivity, etc., enhancing the transport of moisture. Generally, surface cracks cannot be observed until the post-peak phase for cementitious materials under axial compression. Therefore, only the development of inner microcracks is considered for the pre-peak damage process. Moreover, for cementitious materials in service, the stress level is less than 80% of the ultimate compressive strength in most situations, so it is reasonable to assume that the development of microcracks is relatively stable and homogeneous and that the dimensions of microcracks, especially after unloading, are comparable to the pore size.

To consider the effect of axial-compression-induced damage on moisture transport, the basic idea of the dual porous medium model [44] was adapted in the present research; that is, the newly developed microcracks were assumed as another pore network which was different from the original one for undamaged cementitious materials. Therefore, the pore size distribution function, $G_{D}\left(r\right)$, of damaged specimens could be deemed as a superposition of the pore size distribution function, G(r), of undamaged specimens. The pore size distribution function, ΔG(r), of newly developed microcracks and the sorptivity, $S_{oD}$, of damaged specimens could be divided as the sorptivity, $S_{o}$, of undamaged specimens and the sorptivity, $\Delta S_{o}$, of newly developed microcracks. According to Equation (11), the expression of $\Delta S_{o}$ could be written as
(12)$$\Delta S_{o}=\frac{\Delta \Phi }{\tau }\int_{r_{min}}^{r_{max}}\sqrt{\frac{r\sigma }{2\eta }}\Delta G\left(r\right)dr,$$
where ΔΦ is the porosity difference between the damaged and undamaged specimens. Because the precise distribution of ΔG(r) could not be obtained from Equation (12), the damage-representative radius, $r_{D}$, for the damaged mortar or concrete was proposed to remove ΔG(r) by assuming that Equations (12) and (13) gave the same $\Delta S_{o}$, as follows:(13)$$\Delta S_{o}=\frac{\Delta \Phi }{\tau }\sqrt{\frac{r_{D}\sigma }{2\eta }},\;$$

Then, the damage-representative radius, $r_{D}$, could be obtained with the following expression:(14)$$r_{D}=\frac{2\eta }{\sigma }{\left(\frac{\tau \Delta S_{o}}{\Delta \Phi }\right)}^{2}.$$

The damage-representative radii for damaged mortar and concrete were calculated and listed in Table 6. Φ and $\Phi_{D}$ are the mean values of the porosities measured only from the slices in the sorption experiment. Zero was assigned to $r_{D}$ for mortar at the stress levels of 25% and 40% of the ultimate compressive strength due to the appearance of negative numbers in the squared root.

From Table 6, it can be seen that the damage-representative radius, $r_{D}$, is equal to or nearly 0 when the stress level is below 40% of the ultimate compressive strength, but becomes much bigger once the stress level exceeds 55% of the ultimate compressive strength. This is consistent with the observations of axially compressed mortar or concrete. When the stress level is low, the mortar or concrete is elastic, and no residual deformation exists after unloading. However, when stress levels are high enough to cause permanent damage, microcracks develop with the increase in the stress level, and the $r_{D}$ increases. It is also observed that the damage-representative radii of the damaged mortar and concrete are nearly the same when the stress level is 85% of the ultimate compressive strength.

### 4.3. Moisture Diffusivity for Damaged Concrete Based on the Damage-Representative Radius

The moisture diffusivity, $D_{hcp,D}$, for the cement paste specimen subjected to compression-induced damage can be expressed as follows [12]:(15)$$D_{hcp,D}=D_{hcp}+\Delta D_{hcp},$$
where $D_{hcp}$ is the moisture diffusivity for the undamaged cement paste (kg/(m·s)), which reflects the combination of vapor and liquid transport, and can be expressed as
(16)$$\left\{ \begin{array}{c} D_{hcp}=D_{hcp,l}g_{l}+D_{hcp,v}g_{v}\;\;\;\\ D_{hcp,l}=\frac{1}{8}{\left(\int_{r_{min}}^{r_{p}}\frac{r-t_{a}}{\tau }f_{r}\Phi G\left(r\right)dr\right)}^{2}\frac{\rho_{l}^{2}RT}{\eta M_{w}h}\\ D_{hcp,v}=\frac{p_{vs}M_{w}}{\tau RT}\int_{r_{p}}^{r_{max}}\frac{\left(1-f_{r}\right)\Phi D_{va}}{1+l_{m}/\left[2\left(r-t_{a}\right)\right]}G\left(r\right)dr\\ g_{l}=\int_{r_{min}}^{r_{p}}G\left(r\right)dr,g_{v}=\int_{r_{p}}^{r_{max}}G\left(r\right)dr \end{array}\right.$$

Here, $\Delta D_{hcp}$ is the increment of diffusivity compared with the undamaged cement paste (kg/(m·s)), and is related to the damage-representative radius and porosity increment. As previously stated, all the developed microcracks are lumped into a single pore with equivalent sorptivity, so the moisture diffusivity for the damaged cement paste can be calculated as
(17)$$\{\begin{array}{c} \Delta D_{hcp}=\Delta D_{hcp,l}g_{D,l}+\Delta D_{hcp,v}g_{D,v}\\ \Delta D_{hcp,l}=\frac{{\left(r_{D}-t_{a}\right)}^{2}\Delta \phi }{8\eta \tau }\frac{\rho_{l}^{2}TR}{M_{w}h}\\ \Delta D_{hcp,v}=\frac{D_{va}\Delta \phi }{1+l_{m}/[2(r_{D}-r_{a})]}\frac{p_{vs}M_{w}}{\tau RT} \end{array}$$
with
(18)$$\{\begin{array}{c} g_{D,l}=1\\ g_{D,v}=0 \end{array},\;\;r_{D}\leq r_{p}\;\mathrm{or}\;\{\begin{array}{c} g_{D,l}=0\\ g_{D,v}=1 \end{array},\;\;r_{D}>r_{p}$$

Among Equations (16)–(18), $M_{w}$ is the molar mass of water; R is the gas constant; T is the thermodynamic temperature; $r_{p}$ is the actual pore radius; $t_{a}$ is the thickness of the adsorbed layer on the pore wall; $l_{m}$ is the mean free path for the vapor; $p_{vs}$ is the saturated vapor pressure; $D_{va}$ is the vapor diffusivity in the air; $f_{r}$ is the parameter introduced to describe the probability of hysteresis.$\;D_{hcp,l}$ and $D_{hcp,v}$ represent the liquid and vapor diffusivity, while $\Delta D_{hcp,l}$ and $\Delta D_{hcp,v}$ are the damage-caused increment of diffusivity. $g_{l}$ and $g_{v}$ represent the percentages of pore volume taken up by liquid and vapor transport in the undamaged cement paste, while $g_{D,l}$ and $g_{D,v}$ reflect the percentages of the damage-representative radius.

The derivation of Equations (16)–(18) occurred as follows: first, a formula for the thickness of the adsorbed layer was proposed using Kelvin’s equation [6] and Hillerborg’s equation [45]. Then, based on the basic idea of the Barrett–Joyner–Halenda method [4], the pore size distribution functions of cement pastes of different water–cement ratios were calculated using the water adsorption isotherms, which theoretically allow a correlation between the water-cement ratios and the pore size distributions of cement pastes. Finally, with the hysteresis effect under consideration, the moisture diffusivity of cement paste in the wetting-and-drying process was obtained by integrating the calculated pore size distributions. The whole derivation process is extensive; readers are referred to [12] for more details.

The damaged concrete is treated as a two-phase composite material comprised of damaged cement paste and undamaged aggregate, and its moisture diffusivity can be obtained by employing the composite theory [44], as follows:(19)$$D_{h,D}=D_{hcp,D}\left[1+\frac{g_{agg}}{\frac{g_{cp}}{3}+\frac{1}{\left(\frac{D_{hagg}}{D_{hcp,D}}-1\right)}}\right]$$
where $g_{agg}$ and $g_{cp}$ are the volume fraction of aggregate phase and cement paste phase in concrete, respectively, and $D_{hagg}$ is the moisture diffusivity of aggregate (kg/(m·s)). The porosity of aggregates in ordinary concrete is small compared with that of cement paste [46], so that transport inertia can generally be applied to the aggregates.

The moisture diffusivities of concrete subjected to the stress level of 85% of the ultimate compressive strength were calculated with the damage-representative radius, $r_{D}$, of 230.5896 nm and compared with that of undamaged concrete in Figure 11. It can be seen that the moisture diffusivity of the damaged specimen is only slightly higher than that of the undamaged specimen. Similar results can be found in [25]. The measured gas permeability of the damaged concrete specimens is almost the same as that of the undamaged specimens until the applied stress levels exceeded 80% of the ultimate compressive strength at 20 °C or even 90% at 105 °C and 150 °C, whether the measurement was carried out during loading or after loading. More penetration paths are available for external moisture when concrete is damaged, but water vapor transport rather than liquid water transport dominates in $r_{D}$ because, according to the Kelvin equation, capillary condensation for large $r_{D}$ requires very high relative humidity, which cannot be met in normal atmospheric environments. This explains why the marginal increase in the moisture diffusivity for the damaged concrete happens in the low-humidity range. Meanwhile, the porosity difference between the damaged and undamaged concrete is too small to significantly increase the moisture diffusivity. However, it has to be pointed out that the single-tube assumption of the microcrack system is oversimplified, and may not accurately reflect the moisture-transport mechanisms in the microcracks, and both the porosity and sorptivity are measured after unloading, when the microcrack characteristics are believed to be different from those during loading because of elastic recovery; therefore, more work should be carried out in the future to distinguish the pore size distribution function of microcracks from that of the pore system of sound materials, and to obtain more realistic input data from specimens during loading. In these investigations, the question of whether and to what extent mechanical-load-induced microcracks can influence the durability of concrete structures in atmospheric environments can be decided.

## 5. Conclusions

This paper experimentally and analytically investigated the influence of axial-compression-induced damage on the moisture transport of unsaturated mortar and concrete.

It was found from the experiments that both the porosities and sorptivities of the undamaged cement paste, mortar, and concrete increased with the w/c. The porosities of damaged mortar and concrete were slightly bigger than those of undamaged ones. However, the sorptivity was sensitive to the damage caused by axial compression above a threshold, i.e., 40–55% of the ultimate compressive strength, indicating that damage from chloride ion and water transport should be considered for concrete in tidal zones.

The experiments also show that the pore size distributions of mortar and concrete can be accurately derived from that of cement paste; that is, the pore size distribution of cement paste can serve as a reference for that of mortar or concrete with the same w/c when the ITZ is neglected, which can greatly simplify the sample preparation of mortar or concrete for pore size distribution measurements without losing much accuracy.

An analytical model for the sorptivities of damaged mortar and concrete has been deduced and verified, in which damage was simplified to single tubes. The damage-representative radius of the tubes is equal to or nearly zero when the axial compressive stress level is below 40% of the ultimate compressive strength, and increases rapidly once the stress level exceeds 55%. The damage-representative radii of damaged mortar and concrete have marginal differences under the same stress level when the stress level is below 85%.

## Figures and Tables

**Figure 1 materials-15-05498-f001:**
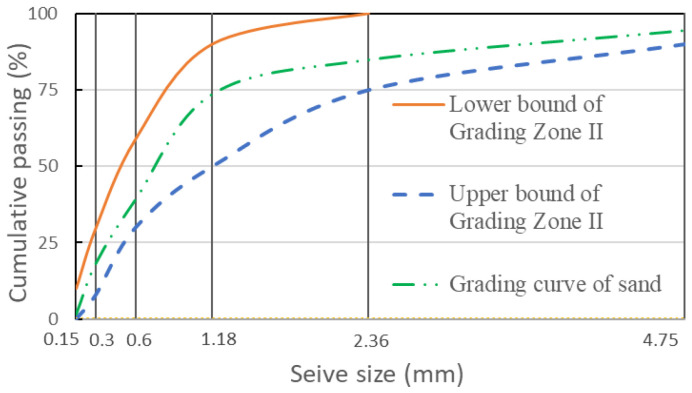
Grading curve of the sand used in the experiments.

**Figure 2 materials-15-05498-f002:**
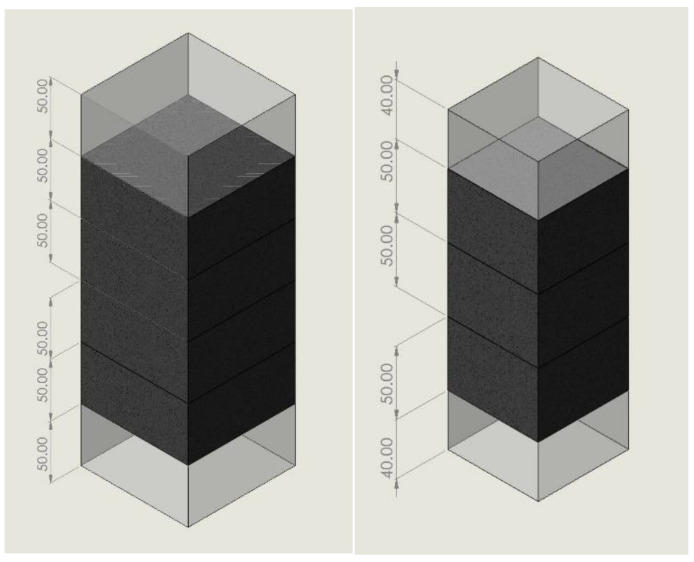
Cutting methods for undamaged concrete (**left**) and cement paste and mortar (**right**) specimens (unit: mm).

**Figure 3 materials-15-05498-f003:**
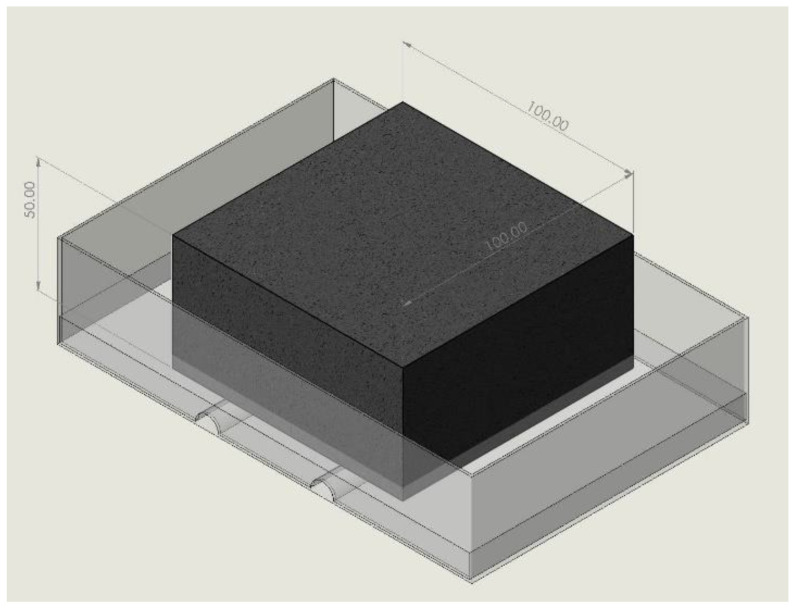
Sorption experiment setup (unit: mm).

**Figure 4 materials-15-05498-f004:**
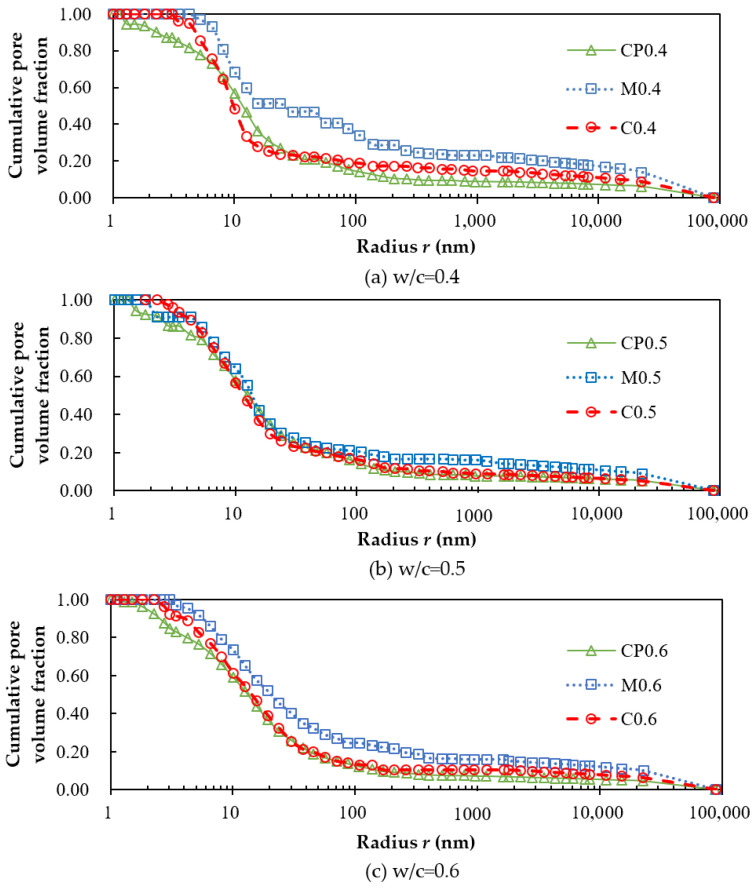
Cumulative pore volume fraction curves of cement paste, mortar, and concrete with the w/cs of: (**a**) 0.4; (**b**) 0.5; (**c**) 0.6.

**Figure 5 materials-15-05498-f005:**
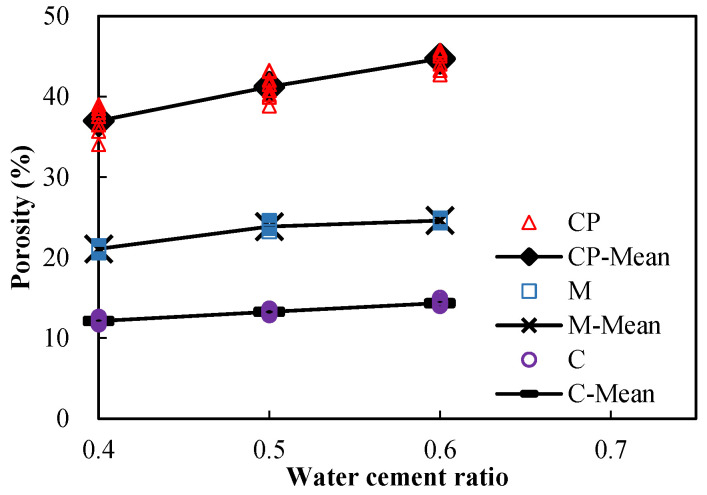
Porosity of undamaged specimens.

**Figure 6 materials-15-05498-f006:**
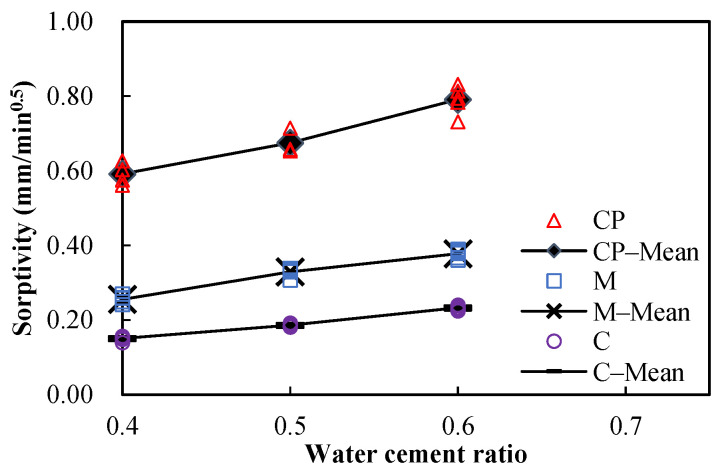
Sorptivity of undamaged specimens.

**Figure 7 materials-15-05498-f007:**
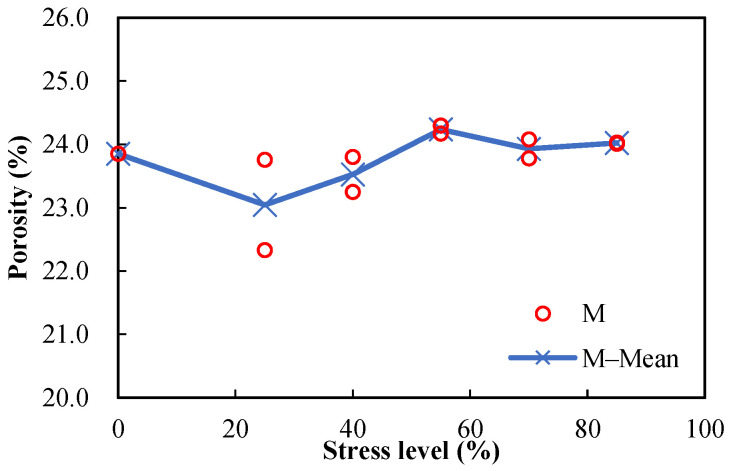
Porosities of damaged mortar specimens (w/c = 0.5) after different compressive stress levels.

**Figure 8 materials-15-05498-f008:**
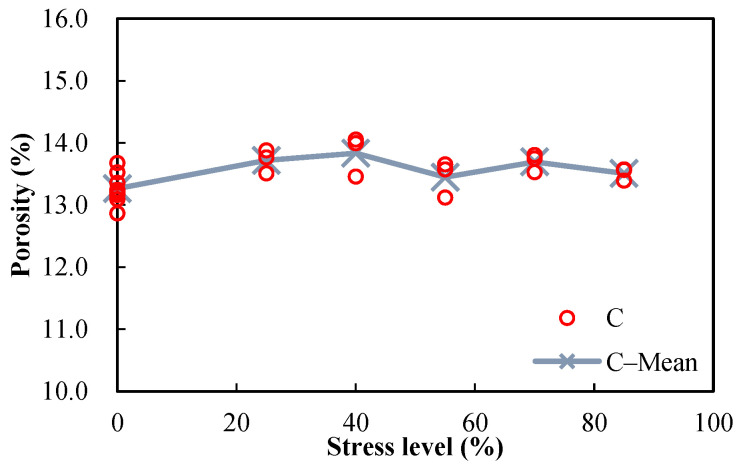
Porosities of damaged concrete specimens (w/c = 0.5) after different compressive stress levels.

**Figure 9 materials-15-05498-f009:**
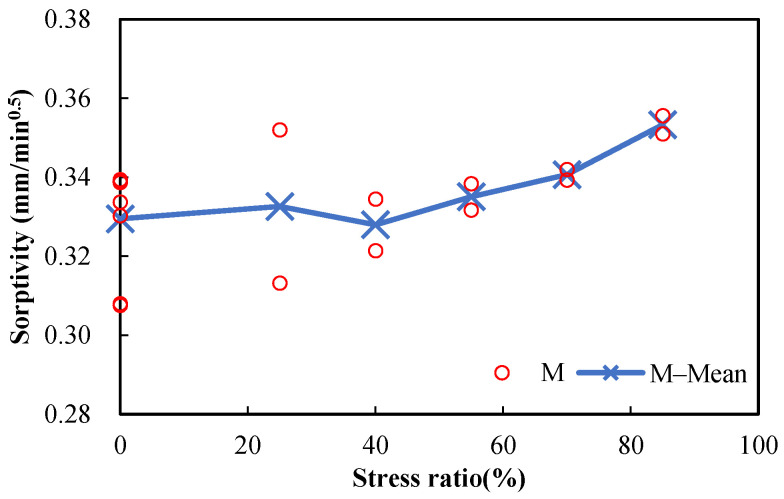
Sorptivities of damaged mortar specimens (w/c = 0.5) after different compressive stress levels.

**Figure 10 materials-15-05498-f010:**
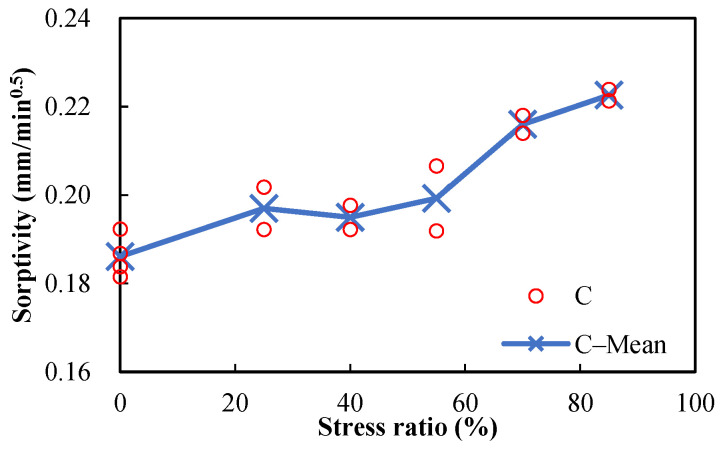
Sorptivities of damaged concrete specimens (w/c = 0.5) after different compressive stress levels.

**Figure 11 materials-15-05498-f011:**
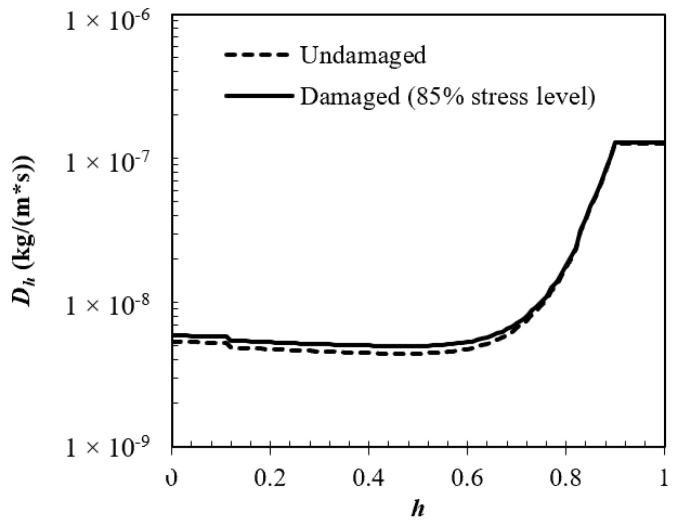
Moisture diffusivity of undamaged and damaged concrete.

**Table 1 materials-15-05498-t001:** Mixture proportions and number of specimens.

Type	No.	Mixture Proportions (kg/m^3^)				Number of Specimens				
		W ^1^	C ^2^	S ^3^	CA ^4^	MIP	AC ^5^	Porosity and Sorptivity		Total
								U ^6^	D ^7^	
Cement paste	CP0.4	552	1379	/	/	1	/	3	/	4
	CP0.5	606	1212	/	/	1	/	3	/	4
	CP0.6	649	1081	/	/	1	/	3	/	4
Mortar	M0.4	331	827	1023	/	1	/	3	/	4
	M0.5	363	727	1023	/	1	3	4	5	13
	M0.6	389	647	1023	/	1	/	3	/	4
Concrete	C0.4	191	478	591	1098	1	/	4	/	5
	C0.5	210	420	591	1098	1	3	3	5	12
	C0.6	225	375	591	1098	1	/	4	/	5

^1^ water; ^2^ cement; ^3^ sand; ^4^ coarse aggregate; ^5^ axial compression; ^6^ undamaged specimens; ^7^ damaged specimens.

**Table 2 materials-15-05498-t002:** Mean porosities and sorptivities, and the calculated porosities of the undamaged specimens.

Type	No.	SM ^1^	PM ^2^ (%)	Cement Paste Phase			Mortar Phase		
				VF ^3^	CP ^4^ (%)	Error ^5^ (%)	VF^3^	CP ^4^ (%)	Error ^5^ (%)
Cement paste	CP0.4	0.5917	36.9925	1	/	/	/	/	/
	CP0.5	0.6753	41.2243	1	/	/	/	/	/
	CP0.6	0.7913	44.7143	1	/	/	/	/	/
Mortar	M0.4	0.2554	21.0931	0.5998	35.1687	−4.93	1	/	/
	M0.5	0.3295	23.8516	0.5993	39.8012	−3.45	1	/	/
	M0.6	0.3779	24.6051	0.5993	41.0585	−8.18	1	/	/
Concrete	C0.4	0.1508	12.1200	0.3463	34.9940	−5.40	0.5733	21.1407	0.23
	C0.5	0.1861	13.2644	0.3465	38.2815	−7.14	0.5735	23.1307	−3.02
	C0.6	0.2331	14.3544	0.3469	41.3825	−7.45	0.5738	25.0151	1.67

^1^ Sorptivity—mean (mm/min^0.5^); ^2^ porosity—mean (%); ^3^ volume fraction; ^4^ calculated porosity (%); ^5^ error = (CP − PM)/PM × 100%, similarly hereinafter.

**Table 3 materials-15-05498-t003:** Measured porosities and sorptivities of damaged mortar specimens (w/c = 0.5).

Stress Level (%)	No.	Porosity (%)			Sorptivity (mm/min^0.5^)		
		Measured	Mean	Error (%)	Measured	Mean	Error (%)
25	* M0.5−11−1	23.7563	23.0435	3.0931	0.3520	0.3326	5.8328
	* M0.5−11−2	22.3308		−3.0931	0.3132		−5.8328
40	* M0.5−14−1	23.2492	23.5260	−1.1767	0.3214	0.3280	−1.9973
	* M0.5−14−2	23.8029		1.1767	0.3345		1.9973
55	* M0.5−17−1	24.1703	24.2343	−0.2642	0.3317	0.3351	−0.9999
	* M0.5−17−2	24.2984		0.2642	0.3384		0.9999
70	* M0.5−10−1	23.7798	23.9300	−0.6277	0.3393	0.3407	−0.3963
	* M0.5−10−2	24.0802		0.6277	0.3420		0.3963
85	* M0.5−23−1	24.0097	24.0194	−0.0406	0.3510	0.3533	−0.6510
	* M0.5−23−2	24.0292		0.0406	0.3556		0.6510

* means the specimen took both porosity and sorption tests.

**Table 4 materials-15-05498-t004:** Measured porosities and sorptivities of damaged concrete specimens (w/c = 0.5).

Stress Level (%)	No.	Porosity (%)			Sorptivity (mm/min^0.5^)		
		Measured	Mean	Error (%)	Measured	Mean	Error (%)
25	* C0.5−11−1	13.8774	13.7172	1.1682	0.2018	0.1970	2.4365
	* C0.5−11−2	13.7666		0.3598	0.1922		−2.4365
	C0.5−11−3	13.5076		−1.5279	/		/
40	* C0.5−13−1	13.4556	13.8339	−2.7347	0.1922	0.1950	−1.4106
	* C0.5−13−2	14.0514		1.5727	0.1977		1.4106
	C0.5−13−3	13.9946		1.1620	/		/
55	* C0.5−15−1	13.6523	13.4471	1.5255	0.1919	0.1993	−3.6888
	* C0.5−15−2	13.5717		0.9262	0.2066		3.6888
	C0.5−15−3	13.1175		−2.4517	/		/
70	* C0.5−17−1	13.7497	13.6934	0.4114	0.2140	0.2160	−0.9259
	* C0.5−17−2	13.8042		0.8096	0.2180		0.9259
	C0.5−17−3	13.5262		−1.2210	/		/
85	* C0.5−20−1	13.5719	13.5080	0.4734	0.2239	0.2226	0.5840
	* C0.5−20−2	13.3907		−0.8683	0.2213		−0.5840
	C0.5−20−3	13.5613		0.3948	/		/

* means the specimen took both porosity and sorption tests.

**Table 5 materials-15-05498-t005:** Comparison of the measured and calculated sorptivities.

Type	w/c	*Φ*–Mean (%)	$S_{0}$ $\left(\mathrm{mm}/\sqrt{min}\right)$		Error (%)
			Measured–Mean	Calculated	
Cement paste	0.4	36.9925	0.5917	0.5261	–11.0884
	0.5	41.2243	0.6753	0.6038	–10.5820
	0.6	44.7143	0.7913	0.7133	–9.8572
Mortar	0.4	21.0931	0.2554	0.2550	–0.1488
	0.5	23.8516	0.3295	0.3523	6.9074
	0.6	24.6051	0.3779	0.3833	1.4184
Concrete	0.4	12.1200	0.1508	0.1619	7.3276
	0.5	13.2644	0.1861	0.1950	4.7985
	0.6	14.3544	0.2331	0.2394	2.6941

**Table 6 materials-15-05498-t006:** Damage-representative radii of damaged mortar and concrete (w/c = 0.5).

Type	*S_o_* $\left(\frac{mm}{\sqrt{min}}\right)$	*Φ* (%)	Stress Level (%)	*S_oD_* $\left(\frac{mm}{\sqrt{min}}\right)$	*Φ_D_* (%)	Δ*S_o_* $\left(\frac{mm}{\sqrt{min}}\right)$	∆*Φ* (%)	*r_D_* (nm)
Mortar	0.3295	23.7586	0	0.3295	23.7586	0	0	0
			25	0.3326	23.0435	0.0031	–0.0072	0
			40	0.3280	23.5260	–0.0015	–0.0023	0
			55	0.3351	24.2343	0.0056	0.0048	3.8592
			70	0.3407	23.9300	0.0112	0.0017	119.1993
			85	0.3533	24.0194	0.0238	0.0026	233.6647
Concrete	0.1861	13.0793	0	0.1861	13.0793	0	0	0
			25	0.1970	13.8220	0.0109	0.0074	6.0256
			40	0.1950	13.7535	0.0089	0.0067	4.8204
			55	0.1993	13.6120	0.0132	0.0053	17.0482
			70	0.2160	13.7770	0.0299	0.0070	51.3804
			85	0.2226	13.4813	0.0365	0.0040	230.5896

## Data Availability

The data presented in this study are available on request from the corresponding author.

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
