# Peer review of "Moisture Transport of Axial-Compression-Damaged Mortar and Concrete in Atmospheric Environment"

_materials, 2022, doi:10.3390/ma15165498_

Round 1
Reviewer 1 Report
Comments for the authors are provided in the attached file.

Reviewer 2 Report
The paper describes new models on moisture diffusivity of concrete and mortar based on the pore size distribution and sorptivity of the undamaged and damaged specimens.
- Please define BJH method (line 69). Maybe the use of the abbreviation is not necessary since it is mentioned only once in the whole text.
- Have you considered building a model based on both your and the same literature available data? To this aim, mathematical modeling should be applied. The accuracy of those relationships and prediction ability of the equations proposed greatly depends not only on the accuracy of the data themselves but also on the quantity of them. The model` accuracy should be checked at least by r2.
- MIP and sorptivity tests should be described in section 2. Please inform on the instrumentation and producers and, in short, about experiment methodology and conditions. If a relevant standard is used to measure the sorptivity, refer to it. Part of the related text from section 3 is to be moved to section 2. Also, describe the machine used to damage the samples.
Reviewer 3 Report
The submitted article, “Moisture Transport of Axial Compression Damaged Mortar and Concrete in Atmospheric Environment” is interesting, original and within the scope of the journal but some changes should be addressed:
1. Please use spaces between figures, tables, equations and text according to journal instructions. It will be more readable.
2. Please provide the information regarding the manufacturer and the original country of the equipments used in materials and methods section.
3. In my opinion the text from line 165 to 173 "Hearn [31]…. used in this paper" must be reformulated such as "In different studies …..[23, 31-33]" or moved in the introduction section".
4. Please do not use numbered list in the conclusion section. Try to express your own conclusions related to each other in order to have fluency.
5. I recommend to check the references and to use the same font and size.
